# Web-Scale Collection of Video Data
# for 4D Animal Reconstruction

**Brian Nlong Zhao**[1,2]     **Jiajun Wu**[1†]     **Shangzhe Wu**[1,3†]

[1]Stanford University     [2]University of Illinois Urbana-Champaign     [3]University of Cambridge

## Abstract

Computer vision for animals holds great promise for wildlife research but often depends on large-scale data, while existing collection methods rely on controlled capture setups. Recent data-driven approaches show the potential of single-view, non-invasive analysis, yet current animal video datasets are limited—offering as few as 2.4K 15-frame clips and lacking key processing for animal-centric 3D/4D tasks. We introduce an automated pipeline that mines YouTube videos and processes them into object-centric clips, along with auxiliary annotations valuable for downstream tasks like pose estimation, tracking, and 3D/4D reconstruction. Using this pipeline, we amass 30K videos (2M frames)—an order of magnitude more than prior works. To demonstrate its utility, we focus on the 4D quadruped animal reconstruction task. To support this task, we present Animal-in-Motion (AiM), a benchmark of 230 manually filtered sequences with 11K frames showcasing clean, diverse animal motions. We evaluate state-of-the-art model-based and model-free methods on Animal-in-Motion, finding that 2D metrics favor the former despite unrealistic 3D shapes, while the latter yields more natural reconstructions but scores lower—revealing a gap in current evaluation. To address this, we enhance a recent model-free approach with sequence-level optimization, establishing the first 4D animal reconstruction baseline. Together, our pipeline, benchmark, and baseline aim to advance large-scale, markerless 4D animal reconstruction and related tasks from in-the-wild videos. Code and datasets are available at `https://github.com/briannlongzhao/Animal-in-Motion`.

## 1   Introduction

The study of animals has long fascinated scientists across fields—from wildlife conservation to biomechanics and robotics. Traditionally, capturing visual data of animal shape and motion requires sophisticated, often expensive, marker-based systems [32, 71]. Modern computer vision techniques offer an alternative approach of purely image-based, markerless motion capture [77, 40, 19, 26]. However, many of these methods depend on multi-view images captured in controlled settings, limiting their applicability to real-world, in-the-wild animal behavior. Recent advancements in tasks such as pose estimation, tracking, and, most challengingly, 3D/4D reconstruction, have enabled efficient analysis of animals from single-view images or videos. Data-driven approaches for animal shape and motion analysis and reconstruction have shown robust performance by leveraging 3D priors, either from scanned models [74, 42, 8, 9] or from optimization over feature correspondences [61, 35, 66]. This opens the door to scalable, non-invasive capture of animal behavior using monocular images and videos collected in the wild. Similar to other areas of machine learning, the progress depends heavily on the availability of large-scale data. Yet, even the largest available animal-centric video datasets remain inadequate, as they contain only 2.4K short clips of 15 frames each, lack object-centric views, and omit crucial data needed for challenging tasks such as 4D animal reconstruction

---

[†]Equal advising.

39th Conference on Neural Information Processing Systems (NeurIPS 2025) Track on Datasets and Benchmarks.

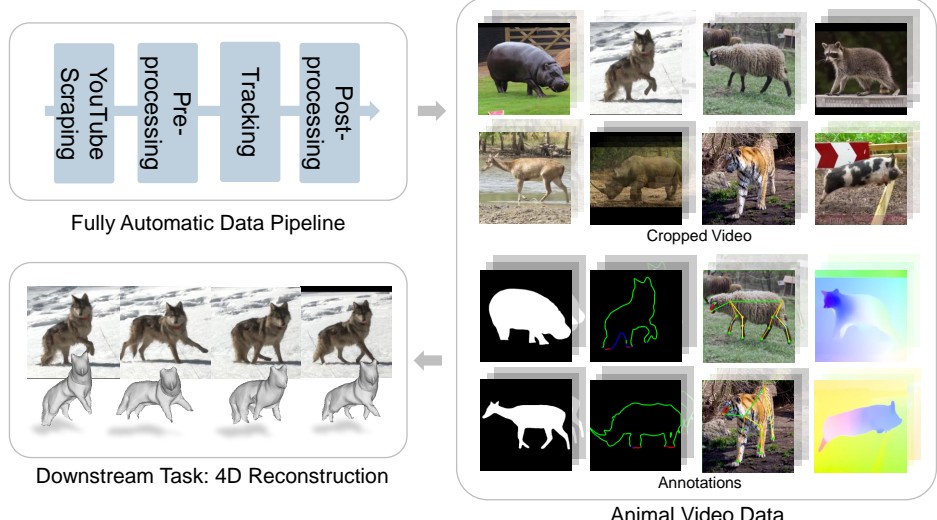

Figure 1: We propose a fully automated data pipeline to collect and process data from scratch, making it ready for downstream tasks such as 4D animal reconstruction.

[68]. The only existing dataset truly suitable for 4D animal reconstruction is even more limited, with only 11 videos in total [8].

In this paper, we introduce a scalable, automated pipeline that enables large-scale video collection and processing for animal shape and motion analysis, with a particular focus on the downstream task of 4D quadruped reconstruction. Our pipeline scrapes raw video from YouTube, exploiting its vast scale and diversity. The videos are then processed into object-centric video clips, along with additional processed features, including instance masks, keypoints, optical flow, and occlusion boundaries, all in a fully automatic manner. These features can be used to aid the downstream task of reconstructing 3D shape and motion of animals, without requiring any explicit 3D annotation. With this pipeline, we obtained an object-centric video dataset of 30K clips, consisting of 2 million frames.

Furthermore, we introduce Animal-in-Motion (AiM), the first benchmark dataset specifically designed for 4D quadruped pose and shape reconstruction. Our dataset contains 230 carefully curated animal motion sequences, totaling 11,061 frames, all collected by our proposed framework. We ensure that each selected sequence has accurate silhouettes and keypoints. We adopt metrics widely used in prior 3D animal reconstruction research. Since no existing methods explicitly target 4D animal reconstruction, we benchmark state-of-the-art 3D approaches—covering both model-based and model-free methods—by evaluating their per-frame performance on Animal-in-Motion. We find that model-based methods often achieve higher scores, yet may produce unrealistic shapes and poses, while model-free methods generate more natural and temporally coherent 3D reconstructions but score lower, revealing a mismatch between evaluation and perceptual quality. This exposes the limitations of 2D-based metrics that are commonly used by the community and underscores the importance of qualitative assessment in 3D and the need for better 3D-aware metrics.

Based on these findings, we also enhance the reconstruction quality of the model-free approach, improving its 2D metrics while retaining natural, consistent 3D poses, shapes, and motion. Specifically, we build upon 3D-Fauna [35], a model-free quadruped reconstruction method that operates in a feed-forward manner across various animal categories. To boost its performance, we incorporate additional keypoint supervision for more precise pose estimation and introduce several optimizations, including smoothness losses, that facilitate effective per-sequence refinement. Our benchmarking and qualitative assessments demonstrate that these modifications yield improvements on all quantitative metrics, while concurrently producing more natural and temporally coherent 3D shapes and motions that better resemble the input videos.

In summary, our contributions are as follows:

- We introduce a unified, automated data pipeline that can collect and process noisy YouTube videos into object-centric clips prepared for downstream tasks such as 4D animal reconstruction.

- We present Animal-in-Motion (Animal-in-Motion), the first benchmark evaluation dataset for 4D quadruped reconstruction.

- We propose 4D-Fauna, a new model-free 4D animal reconstruction baseline that adapts 3D-Fauna with extra guidance and losses to enable more accurate reconstruction on video.

- We present analyses of the benchmarking results for current model-based and model-free approaches, revealing gaps in current metrics and suggesting directions for future evaluation design.

## 2 Related Works

### 2.1 Animal Reconstruction from Image

The task of 4D animal reconstruction extends from 3D animal reconstruction, which focuses on estimating 3D pose and shape of an animal from a 2D image, and 4D reconstruction results can be obtained by applying 3D reconstruction methods independently to each video frame. Numerous studies have explored the 3d animal reconstruction task, primarily using either a model-based or model-free approach. Typical model-based approaches include ABM [4], SMAL [74], and their variations and extensions [58, 75, 76, 9, 48, 49, 31, 8, 38]. These methods start with a prior 3D mesh and optimize a set of predefined pose and deformation parameters to fit the 2D ground truth labels, such as silhouettes and keypoints. These approaches guarantee a consistent 3D shape, however, the predefined shape covers only a limited number of animal categories, and the deformation space is constrained. On the other hand, model-free approaches, including [29, 30, 66, 9, 61, 60, 69, 24, 33, 2, 16, 65], learn a category-specific canonical shape from large-scale image datasets containing different instances of the same category. General-domain 4D reconstruction methods [16, 65] can also be applied to animals, but their reconstructed results lack part- or joint-level information, making them less flexible than animal-specific reconstruction approaches. At test time, they predict instance-specific pose and deformation parameters in a feed-forward manner. Additionally, 3D-Fauna [35] learns a generalizable prior shape bank from pan-category image data, aiming to predict category-agnostic prior shapes at test time. Model-free methods generally accommodate a more diverse range of shapes, however, their feed-forward nature at test time limits the accuracy of shape and pose reconstruction. In this work, we propose a new baseline method that combines the advantages of model-based and model-free approaches by directly optimizing per-frame parameters of a pretrained model-free model at test time, incorporating additional geometric and temporal loss terms on video data.

### 2.2 Web Data Collection

As machine learning models continue to scale rapidly, the demand for larger-scale datasets has become increasingly critical for effective training. The most effective approach to data collection is leveraging the vast amount of information available on the web, processing it into structured datasets tailored for specific tasks. The training of state-of-the-art language models [3, 22, 52] often relies on large-scale text datasets scraped from the web, such as Common Crawl. Large-scale image datasets sourced from the web [50, 17] have also served as foundational resources for modern computer vision research. In the domain of video, many datasets obtain video data from online video-sharing platforms such as YouTube, Flickr, and Tumblr. Datasets such as ActivityNet [20], Kinetics [25], and YouTube-8M [1] utilize online videos as sources for classification tasks. Other datasets, such as HD-VG-130M [57], HD-VILA-100M [64], TGIF [34], MiraData [23], HowTo100M [39], WebVid-2M [5], collect text-video paired data from online sources for tasks including video captioning, retrieval, and generation. For more specialized tasks, however, specific approaches or processing pipelines are required to process and filter video data, often incorporating human annotations. Instructional datasets such as COIN [53] and IKEA Video Manuals [37] require human annotators to use annotation tools to create labels for the data. VoxConverse [15] and VGG-Sound [14] propose carefully designed automatic pipelines for audio-visual data annotation, only requiring minimal human effort for verification. Similarly, for the specific task of 4D animal reconstruction, we aim to develop a scalable automated pipeline for data preparation and processing, eliminating the need for human annotation while minimizing human effort for verification.

## 2.3 Animal Datasets

Many vision datasets focus specifically on animals, primarily designed for the task of animal pose estimation, where models are expected to predict the spatial configuration of animals given an image or other modality data. Some datasets focus on specific animal categories such as dogs, birds, and monkeys [18, 27, 56, 70]. Others encompass a diverse range of animal species but are limited to images without video data [72, 11, 41, 62]. There are some video-based animal datasets, such as BADJA [8] and APT-36K [68], EgoPet [6]. However, they suffer from two key limitations. First, they remain small in scale—the largest among them, APT-36K, contains only 2.4k videos, each with 15 frames. Second, these datasets consist of unprocessed in-the-wild videos, where typical frames may include multiple overlapping animals, lack center-cropping, and omit essential data such as masks. As a result, these datasets are not fully prepared for the task of 4D animal reconstruction, making them unsuitable for direct use by 4D animal reconstruction methods. To develop a scalable solution that produces ready-to-use data for animal reconstruction, we leverage large-scale online video data and design an automatic pipeline to collect and process structured datasets ready for 4D animal reconstruction task.

## 3  Data Collection

We propose a multi-stage data engine that can automatically collect and process data for 4D animal reconstruction task. The process follows a pipeline that searches for video candidates, applies tracking algorithms for object-centric video crops, filters out unwanted tracks, and leverages off-the-shelf pretrained vision models to extract all necessary image features for animal reconstruction. At the core of our data engine lies a database that stores metadata of intermediate results collected or processed at different stages, enabling parallel execution of multiple processes running same or different stages. An overview of our data engine is shown in Figure 2.

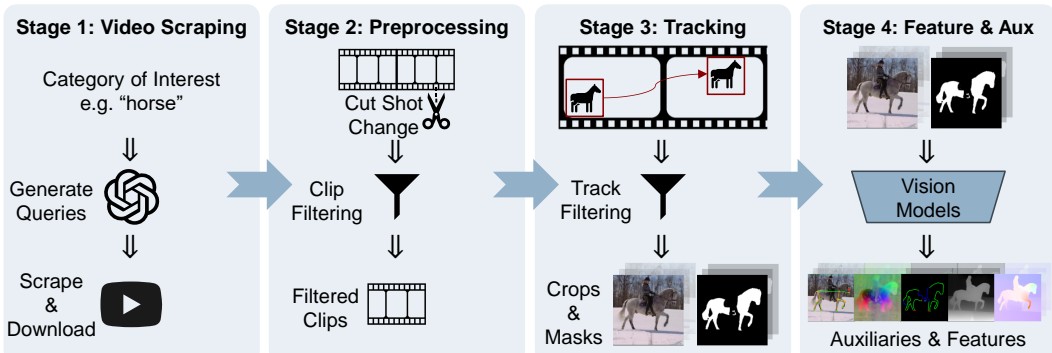

Figure 2: Overview of our proposed data pipeline. Data is automatically scraped from YouTube in stage 1, and then preprocessed into clips in stage 2. Stage 3 detects and tracks the instances, and the final stage extracts additional image features for 4D animal reconstruction.

## 3.1  Raw Video Collection

In this stage, we scrape and download raw videos from YouTube. We start with an arbitrary animal category or family, for example, horse, and leverage GPT to generate text search queries. To make search queries as diverse as possible, we first ask GPT to generate a set of more specific sub-category breeds, for example, *Clydesdale* and *Mustang*. Separately, GPT is asked to generate a set of context phrases that are related to the category of interests, for instance, *racing competition* and *in a farm* for horse. Finally, we randomly combine two sets to form a list of diverse search texts to query YouTube for raw videos. Our downloading pipeline is implemented based on `Selenium Webdriver` [51] for querying and retrieving video ID results and `pytube` [10] for downloading the videos.

## 3.2 Video Preprocessing

This stage aims to preprocess and prefilter the downloaded raw videos so that they are prepared for object tracking in the next stage. The final data we want should be object-centric crops of animals, but raw videos are typically noisy with many frames that do not have any animal of interest presented in the frame. We therefore split the video into video clips based on shot changes using `PySceneDetect` [13], which detects large values of weighted average pixel change across frames. Splitting videos by shot change also helps tracking as we observe that tracking algorithms may falsely associate objects in different shots. Next, we apply CLIP [45] and compute an average CLIPScore [21] between a randomly sampled batch of frames from the video clip and a text caption, for example, *a photo of a horse*, and discard video clips with low CLIPScore. This step filters out video clips that do not clearly depict the target animals, thereby eliminating the need for further tracking. All video clips are downsampled to 10 frames per second to enhance processing efficiency in later stages.

## 3.3 Animal Tracking

To obtain object-centric video cropping of an animal across frames, it is essential to track the same instance consistently over multiple frames. We employ Grounded-SAM-2 [47], which utilizes GroundingDINO [36] to detect object bounding boxes in each frame, serving as prompt inputs for SAM-2 [46] tracking. An iterative grounding-tracking process is applied to enable long-term tracking while also allowing the detection and tracking of newly appearing objects. After tracking, we obtain a set of object track proposals represented by their corresponding mask silhouettes from a video clip. We filter out any tracks that are potentially unsuitable for animal reconstruction task by applying the following filters and postprocessings.

**Overlapping Instances.** When multiple animals are present in a frame, off-the-shelf keypoint estimators may become confused and incorrectly assign keypoints to different instances, especially when significant overlap occurs. To mitigate this, we remove frames from tracks where two or more animals overlap substantially. We achieve this by thresholding the Intersection over Union (IoU) between each pair of animals in the same frame and removing both mask silhouettes from the tracks if their IoU exceeds the threshold.

**Low Resolution Instances.** If the animal is too small in the frame, subsequent operations such as keypoints and feature extraction may have degraded performance due to low resolution after resizing. Therefore we discard any frames from the track where the bounding box area of the animal is less than 1/4 of the final crop size, *e.g.* $256 \times 256$ if the final crop size is $512 \times 512$.

**Truncated Instances.** In many cases, an animal's full body is not visible within the video frame. Since animal reconstruction methods rely on mask silhouettes as shape supervision, truncated silhouettes can lead to inaccurate reconstructions with unnatural poses and shapes. We remove frames from the tracks if the bounding box is too close to the frame border, as these typically indicate a truncated animal.

**Inconsistent Tracks.** Tracking algorithms may fail when videos contain ambiguous cases or unnatural artifacts. A common failure occurs when multiple animals with similar appearances are present, causing the algorithm to switch identities and track different animals inconsistently. Another failure case arises from video fading effects, which are difficult to detect using shot detection algorithms in earlier stages. In some instances, the tracking algorithm may fail to stop even after a shot change or fade-out, continuing to track a different object or background in the new shot. To mitigate these issues, we apply a threshold on the bounding box IoU between adjacent frames of the same track and remove all frames following a detected low IoU.

**Temporal Postprocessing.** At this stage, some unqualified frames have been filtered from the tracking results, creating discontinuities. To address this, we apply a post-processing step based on predefined parameters for minimal track length, maximal track length, and the allowed gap within a track. By iterating through all frames in a track, we identify gaps exceeding the allowed threshold or instances where the track reaches the maximal length; in such cases, the subsequent frames are split into a new track. If a gap falls within the allowed threshold, we resegment missing mask silhouette

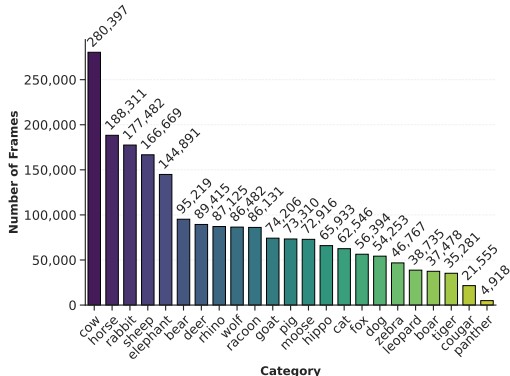
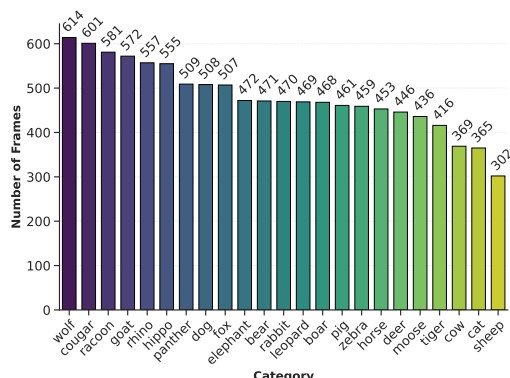

Figure 3: A detailed breakdown of the number of frames collected for each animal category in full dataset.

Figure 4: A detailed breakdown of the number of frames collected in benchmark data for each animal category.

using SAM-2, interpolating bounding boxes from both sides as input prompts. Any tracks shorter than the minimal track length are discarded.

**Object-centric Cropping.** To obtain the final object-centric video crops for animal reconstruction, we generate square crop boxes centered on the bounding box of the animal in each frame. The size of each crop box is determined by a predefined ratio relative to the mask area. We further apply moving average smoothing to the crop boxes before cropping and resizing all frames to a standardized size. As a final filtering step, we randomly select a cropped RGB image for each track and input it into GPT to identify and remove instances of false detection or heavily occluded animals.

## 3.4 Feature Extraction

Animal reconstruction methods typically rely on additional preprocessed features beyond RGB images to guide optimization. Most model-based approaches optimize shape and pose parameters to fit animal silhouettes and keypoints. Some methods [28] may also use optical flow as additional guidance. Model-free methods further incorporate precomputed image features such as DINO features [69, 61, 35]. While animal silhouettes are already obtained in a previous stage, our data pipeline modularly integrates off-the-shelf vision models to automatically infer animal keypoints, PCA DINO features, optical flow, and depth. Specifically, we integrate ViTPose++ [63] for animal keypoints estimation, DINOv2 [43] for image feature, SEA-RAFT [59] for optical flow estimation, and Depth Anything V2 [67] for depth estimation. Additionally, we compute occlusion boundaries for each animal crop based on the estimated depth and mask silhouette. Specifically, we extract depth values at the dilated and eroded mask boundaries, respectively. For each pixel on the original mask boundary, we calculate the depth difference between the nearest pixel on the dilated boundary and the nearest pixel on the eroded boundary. This depth difference helps determine whether the pixels outside the animal silhouette belong to the foreground, indicating occlusion, or the background, indicating no occlusion at that region. Using the estimated optical flow and occlusion boundaries, we can optionally further filter the processed data to retain instances with greater motion and minimal occlusion.

## 3.5 Dataset Statistics

Using the proposed automated data pipeline, we can efficiently collect a large volume of structured video data suitable for 4D animal reconstruction. We have successfully collected and processed 29,979 animal video data, totaling 2,046,414 frames. Specifically, we begin with 23 common animal categories as input to our data engine, and we present category statistics in Figure 9. A typical data sample collected is shown in Figure 5.

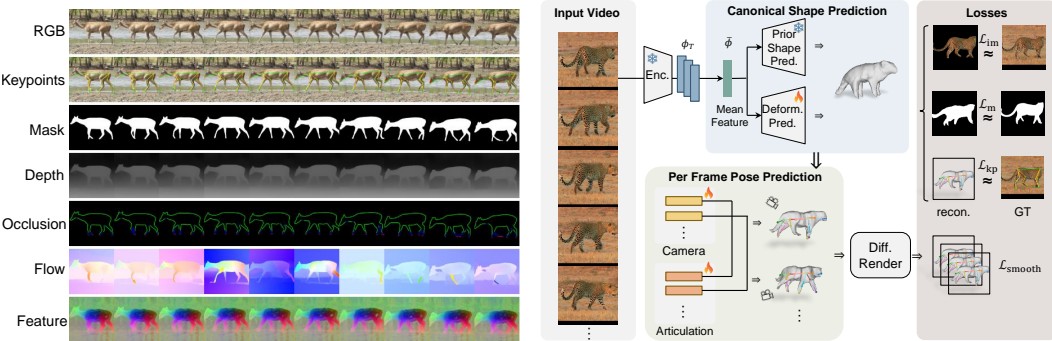

Figure 5: Visualization of a collected data sample. Each video is annotated per frame with keypoints, masks, depth maps, occlusion boundaries, optical flow, and DINO features.

Figure 6: Overview of 4D-Fauna. We adapt 3D-Fauna [35] to enhance its capability for direct sequence optimization.

## 4 4D Animal Reconstruction

Using our proposed data engine, we can generate large-scale datasets for 4D animal reconstruction. To demonstrate its effectiveness, we establish a benchmark dataset, Animal-in-Motion, for evaluating 4D animal reconstruction methods. In this section, we detail the task of 4D animal reconstruction, the construction process of the benchmark dataset, and the evaluation metrics. Additionally, we propose a novel baseline method, 4D-Fauna, that adapts 3D-Fauna [35] with additional loss terms, enhancing its suitability for direct sequence optimization while combining the strengths of both traditional model-based and model-free approaches. Finally, we present benchmarking results comparing a typical model-based method, a model-free method, and our proposed baseline method.

### 4.1 Task Formulation

Similar to the 3D animal reconstruction task, which aims to estimate an animal's 3D pose and shape from a single image, the 4D animal reconstruction task seeks to estimate a sequence of 3D poses and shapes from a sequence of frames of the same animal. Beside RGB image input, typical methods also requires other 2D auxiliary data as guidance, such as mask silhouettes and 2D keypoints, obtained either from manual labeling or from pretrained vision models. Formally, given an RGB video input $\mathcal{V}_T = \{v_t\}_{t=1}^T \in \mathbb{R}^{T \times 3 \times H \times W}$ of an animal, along with any required auxiliary input $\mathcal{A}_T = \{a_t\}_{t=1}^T$, in $\{0,1\}^{T \times H \times W}$ in the case of 2D mask or in $\mathbb{R}^{T \times K \times 2}$ in the case of 2D keypoints for instance, a function $f_\theta : \{\mathcal{V}, [\mathcal{A}]\} \mapsto \mathcal{S}$ is expected to output a sequence of posed 3D shapes $\mathcal{S}_T = \{s_t\}_{t=1}^T$ that naturally resembles the shape and pose sequence shown in the input video $\mathcal{V}_T$, where $T$ is number of frames in sequence, $H$ and $W$ are spatial dimensions of the frames, $K$ is number of defined keypoints. Function $f_\theta$ can operate either as a feed-forward model using pretrained parameter $\theta$, or by optimizing $\theta$ at test time. Since different methods operate differently—some requiring large-scale training data [35] while others only perform test-time optimization [74, 7]—our benchmark dataset is designed for evaluation only to ensure fair comparisons.

### 4.2 Benchmark Dataset

Using the data pipeline proposed in Section 3, we can effortlessly collect large-scale data for the 4D animal reconstruction task from scratch. However, to establish a benchmark, it is crucial to ensure the accuracy of all annotations. This is achieved through human validation, requiring minimal effort from annotators, who simply accept or reject a data sample by reviewing three auxiliary visualizations generated by the data pipeline: the RGB video, RGB video applied with per frame mask silhouette, and RGB video overlaid with per frame keypoints visualization. The criteria for accepting a data sample are as follows:

- The RGB video does not exhibit heavy occlusion of the animal by other objects, particularly on the legs, though self-occlusion is allowed.

- The RGB video displays recognizable and smooth motion in animal body parts.
- The RGB video displays smooth camera movement.
- The RGB video applied with per frame mask silhouettes correctly segments the animal without significant missing body parts.
- The RGB video overlayed with per frame keypoints accurately and smoothly approximates the animal's joint positions across frames.

As a result, we curate 10 videos per category, yielding a total of 230 videos comprising 11,061 frames. A detailed breakdown of the frame statistics is presented in Figure 4.

## 4.3 Metrics

It is non-trivial to evaluate a 2D-to-3D lifiting task since there is no 3D ground truth data. However, literatures have used different proxies to evaluate.

**Silhoutte Intersetion-over-union (IoU).** We follow previous works [9, 76, 4, 58, 49, 69] to employ silhouette intersection-over-union (IoU). Silhoutte IoU measure the IoU between the ground truth silhouette mask and the silhouette mask rendered by the reconstructed 3D shape. Although a high 2D IoU does not necessarily correspond to a natural 3D shape due to potential ambiguities, a low IoU reliably indicates that the reconstruction is underperforming.

**Percentage of Correct Keypoint (PCK).** Following previous works [24, 33, 30, 60, 61, 35], we use the Percentage of Correct Keypoints (PCK) metric, which measures the percentage of projected keypoints that fall within a fixed multiple of a normalizing distance threshold. Studies have defined different distance thresholds. Following [9, 31, 8], we use the square root of the ground-truth mask silhouette area as the normalizing distance threshold.

**Keypoint Transfer (KT).** Since no ground-truth 3D keypoint or shape annotations exist, previous works [61, 35, 29, 30, 69, 33, 8, 28] use Keypoint Transfer (KT) as a proxy for evaluating reconstructed 3D shapes. Specifically, a set of ground-truth 2D keypoints from a source image is projected onto the reconstructed 3D shape surface to establish a mapping with surface vertices. The corresponding vertices are then reprojected from the 3D shape onto a target image with novel view and pose. PCK is computed using the reprojected keypoints and the ground-truth keypoints in the target image. A well-reconstructed 3D shape should exhibit consistency, producing low errors after undergoing this 2D-to-3D-to-2D mapping.

**Mean Per-Joint Velocity Error (MPJVE).** As our work is the first to specifically focus on the task of 4D animal reconstruction, there are no established metrics for evaluating reconstructed motion in the temporal dimension. Following related works in human motion estimation [44, 54, 73], we adopt Mean Per-Joint Velocity Error (MPJVE) to quantify the discrepancy in joint velocity within the projected, normalized pixel space. Specifically, for each joint across two consecutive frames, we compute the magnitude of the vector difference between the ground-truth velocity and the predicted velocity. The final MPJVE is obtained by averaging the error over all joints and all frames.

## 4.4 4D-Fauna

We propose 4D-Fauna, a new baseline for 4D animal reconstruction, which adapts 3D-Fauna [35], a model-free reconstruction approach designed for pan-category quadruped 3D reconstruction. Figure 6 gives an overview of our method.

**Preliminary.** 3D-Fauna builds upon MagicPony [61], which learns a prior shape for a specific animal category from diverse images of that category by leveraging self-supervised DINO-ViT [12] features. It then applies instance-specific predicted parameters, such as deformation and articulation, for inverse rendering supervision. Building on this, 3D-Fauna introduces a learnable prior shape bank, which functions as a dictionary of features capable of dynamically combining basis shapes during training and inference to generate diverse instance-specific prior 3D shapes. As a result, 3D-Fauna removes the constraint of training and inference on a single category, enabling it to learn a rich prior shape bank from pan-category images and produce diverse prior shapes at inference time.

| Method | IoU↑ | PCK@0.1↑ | PCK@0.05↑ | KT-PCK@0.1↑ | KT-PCK@0.05↑ | MPJVE↓ |
|--------|------|----------|-----------|-------------|--------------|--------|
| SMALify [7] | 0.867 | 0.954 | 0.787 | 0.623 | 0.372 | 0.023 |
| AniMer [38] | 0.677 | 0.537 | 0.199 | 0.566 | 0.256 | 0.038 |
| 3D-Fauna [35] | 0.670 | 0.470 | 0.177 | 0.329 | 0.130 | 0.058 |
| 4D-Fauna | 0.814 | 0.664 | 0.317 | 0.418 | 0.193 | 0.044 |

Table 1: Benchmark results comparison of different 4D animal reconstruction methods. blue represents model-based approach and red indicates model-free approach.

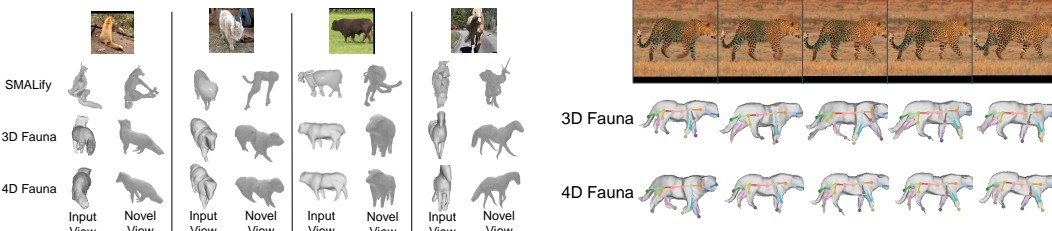

Figure 7: Comparison of 3D reconstruction results highlighting failure cases of SMALify.

Figure 8: Comparison of 4D reconstruction results highlighting failure cases of 3D-Fauna.

**Sequence Optimization.** 3D-Fauna operates in a feed-forward manner, differing from traditional model-based approaches that iteratively optimize a predefined shape to fit a single image through inverse rendering. Consequently, it may yield suboptimal shape and pose fitting compared to model-based methods that overfit to the 2D supervision. To address this, we leverage the pretrained 3D-Fauna and perform per-sequence optimization for 4D reconstruction. However, directly optimizing on a sequence presents several challenges as follows.

**Pose Ambiguity.** Since 3D-Fauna relies solely on 2D mask supervision for the entire projected silhouette without any part-level constraints, the reconstructed results often fail to accurately preserve the correct leg ordering in the image. This error becomes more pronounced in video reconstruction, leading to unnatural gait cycles where the legs fail to switch properly when the animal is walking or running, particularly in side-view perspectives. To address this issue, we explicitly incorporate 2D keypoint annotations as part-level supervision during sequence optimization, similar to how keypoint reprojection loss is utilized in model-based approaches.

**Temporal Smoothness.** We apply temporal smoothness loss terms on both the camera pose and animal pose. Specifically, we regularize the magnitude of change in camera pose parameters and velocity of animal pose articulation.

**Efficient Overfitting.** 3D-Fauna uses neural network predictors to predict camera pose and articulation parameters from input image features. To efficiently overfit camera pose and articulation for the sequence, we directly optimize the camera pose and articulation parameters for each frame, taking the output from pretrained neural network predictors as initialization.

### 4.5 Results and Analysis

We show benchmark results of 4D-Fauna, 3D-Fauna [35], SMALify [7], which is a model-base reconstruction approach that implements [8] and [9], and AniMer [38], which is a model-base feedforward reconstruction method. The quantitative results are reported in Table 1.

As a model-based method, SMALify achieves the best results across all metrics. This is because model-based methods explicitly optimize pose and shape deformation to align with 2D ground truth, leading to superior performance on 2D metrics. However, this optimization process is not inherently 3D-aware—i.e., it does not learn general animal pose and shape representations from diverse data. As a result, when fitting to 2D supervision, the method often produces unnatural poses and shapes due to ambiguities in the depth dimension. Some failure cases of SMALify are illustrated in Figure 7.

The first column from the left illustrates an incorrect pose prediction. Since both the correct and incorrect poses can perfectly fit the 2D mask silhouette and keypoints, the model fails to differentiate between them in this case. The second column demonstrates that in the depth dimension, SMALify may arbitrarily elongate body parts, as such distortions are not apparent in the projected 2D shape. The third column presents a similar failure due to depth ambiguity, where the legs unnaturally bend sideways in 3D space to conform to the 2D supervision. The last column highlights how SMALify drastically deforms the shape into an unnatural configuration to perfectly fit a frontal-view image. In contrast, both 3D-Fauna and 4D-Fauna effectively infer plausible 3D shapes and natural poses. Moreover, 4D-Fauna generally achieves more accurate camera and animal pose fitting than 3D-Fauna, thanks to further sequence-level optimization.

From quantitative results and qualitative assements shown in Figures 7 and 8 4D-Fauna achieves better performance on all metrics than 3D-Fauna while maintaining plausible and natural 3D shape and pose. Specifically, further optimization using mask silhouette supervision contributes to a higher IoU, promoting improved per-frame pose estimation and more accurate shape deformation for individual instances. Although the joint definitions are not fully aligned, direct keypoint supervision on joints with overlapping definitions is sufficient to reconstruct a more accurate animal pose, leading to a higher PCK score. Furthermore, improved camera and animal pose estimation together enhance Keypoint Transfer accuracy in the 2D-to-3D-to-2D mapping process. Additionally, loss terms for motion smoothness help reduce jitter and sudden large movements, producing a more stable and realistic motion that closely resembles the ground truth. A comparison of 4D reconstruction result is shown in Figure 8. Comparing the two model-free approaches, 3D-Fauna exhibits sudden leg switching between frames 2 and 4 in Figure 8, whereas 4D-Fauna successfully resolves this issue, highlighting the necessity of further sequence optimization with keypoint supervision and smoothness loss terms.

## 5  Conclusion

We present a fully automated, scalable data pipeline for 4D quadruped animal reconstruction. We introduce Animal-in-Motion, the first benchmark for 4D animal shape and pose estimation, and establish a thorough evaluation framework for existing 3D reconstruction approaches. Additionally, we propose 4D-Fauna, a baseline that boosts model-free reconstruction accuracy. Our results show the pipeline's ability to generate high-quality data, also highlighting the importance of 3D-aware evaluation and visualization on the animal reconstruction task, opening avenues for further advances in shape and motion understanding of animals.

**Limitation.**  While our pipeline significantly reduces the human effort required for large-scale animal video collection and annotation, the automatically processed data is not perfectly clean and still requires manual validation for reliable benchmarking. Our benchmark dataset, though curated, relies on 2D projection-based metrics, which are limited by inherent view ambiguities and do not fully capture 3D reconstruction quality—highlighting the need for more robust, 3D-aware evaluation metrics. Finally, our baseline builds on an existing model-free method with temporal refinements, but it demonstrates only limited understanding of temporal coherence; future approaches may benefit from more expressive paradigms, such as autoregressive models, to better capture inter-frame dynamics.

**Accessibility.**  The source code and instructions to download dataset are available on GitHub: `https://github.com/briannlongzhao/Animal-in-Motion`.

Our dataset includes annotations derived from publicly available YouTube videos. We acknowledge that YouTube content is subject to copyright protection and governed by YouTube's Terms of Service. To respect these terms and mitigate copyright and privacy concerns, we do not release the original RGB video frames. Instead, we publicly release only derived data, such as mask, depth, keypoints, *etc.*, which do not contain any raw video content. All derived data is non-identifiable and used solely for research purposes. We also provide scripts to re-derive necessary data and visualizations locally, ensuring reproducibility for research purpose.

## Acknowledgments and Disclosure of Funding

This work is in part supported by ONR MURI N00014-22-1-2740, NSF RI #2211258, and #2338203.

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

**Appendix**



# A Data Pipeline Implementation Detail

## A.1 Database

We incorporate SQLite, a lightweight local file-based database system, to store metadata generated during processing and enable parallel execution of multiple processes across the same or different stages. Specifically, each intermediate result after Stage 1 and Stage 2 contains a status field that records whether a video or clip is unprocessed, being processed, completed, or discarded. These statuses are dynamically checked and updated by the processing pipeline, ensuring efficient parallel execution on large-scale data. Additionally, we store metadata such as video titles, query phrases, and keywords, which may facilitate the development of multimodal animal motion studies, such as motion retrieval and generation.

## A.2 Video Scraping

We leverage GPT for automatic search query generation. To make search queries as diverse as possible, we first ask GPT to generate a set of more specific sub-category breeds, for example, *Clydesdale* and *Mustang*. Separately, GPT is asked to generate a set of context phrases that are related to the category of interests, for instance, *racing competition* and *in a farm* for horse. Finally, we randomly combine two sets to form a list of diverse search texts to query YouTube for raw videos. Specifically, given a category name of an animal, we use the following prompt to generate diverse sub-category or breed, where *n* is set to 10, and *category* is the name of category of interests, *e.g. horse*:

> *List {n} types of {category}. Only show the list in python list format without using a code block.*

Similarly, we generate query phrases with the following prompt, with *n* set to 10 as well:

> *List {n} search phrases or autocompletions for searching {category} videos on a video sharing website. Assume user already input the word category, only show the trailing phrases. Only show the list in python list format without using a code block.*

We then combine them in a set-product manner to generate search queries for YouTube.

Since the YouTube API imposes rate and usage limits, we employ a web automation framework to simulate human searches via a web browser, which bypasses these restrictions.

After this stage, each downloaded video is stored in the database with a unique YouTube video ID.

Our downloading pipeline is implemented based on `Selenium Webdriver` [51] for querying and retrieving video ID results and `pytube` [10] for downloading the videos.

## A.3 Preprocessing

This stage preprocesses raw videos for object tracking, aiming to create animal-centric crops. Raw videos are filtered to remove frames lacking the target animal.

At this stage, downloaded videos are first retrieved from the database. The videos are then cut into clips based on shot changes using `pyscenedetect` [13].

Specifically, we compute the average pixel difference between consecutive frames in the HSV color space and set a threshold of 25 to determine shot boundaries. Clips shorter than 30 frames are discarded. While this algorithm is effective in most cases, it fails to detect fading effects, which we address in Stage 3. Using a similar pixel-difference-based approach, we also remove clips consisting solely of still frames with no pixel changes. Additionally, we compute an average CLIPScore for each clip by randomly sampling 10 frames and comparing them against the prompt: "A photo of category." All video clips are downsampled to 10 frames per second to enhance processing efficiency in later

stages. Similar to the first stage, the status of the source video and processed clips is dynamically monitored and updated throughout this process.

## A.4  Tracking

At this stage, processed clips are retrieved from the database, and tracking with segmentation is performed.

Specifically, we apply grounding results on the first frame as a visual prompt to SAM, then track for 50 consecutive frames. This process is iterated for the next 50 frames, using location-based association to link tracks between the end of the current interval and the beginning of the next.

This enables long-term tracking while also allowing the detection and tracking of newly appearing objects.

After obtaining the initial tracking results with segmentation, we sequentially apply the filtering steps.

**Overlapping Instances.**   When multiple animals are present in a frame, off-the-shelf keypoint estimators may become confused and incorrectly assign keypoints to different instances, especially when significant overlap occurs. To mitigate this, we remove frames from tracks where two or more animals overlap substantially. We achieve this by thresholding the Intersection over Union (IoU) between each pair of animals in the same frame and removing both mask silhouettes from the tracks if their IoU exceeds the threshold.

**Low Resolution Instances.**   If the animal is too small in the frame, subsequent operations such as keypoints and feature extraction may have degraded performance due to low resolution after resizing. Therefore we discard any frames from the track where the bounding box area of the animal is less than 1/4 of the final crop size, *e.g.* $256 \times 256$ if the final crop size is $512 \times 512$.

**Truncated Instances.**   In many cases, an animal's full body is not visible within the video frame. Since animal reconstruction methods rely on mask silhouettes as shape supervision, truncated silhouettes can lead to inaccurate reconstructions with unnatural poses and shapes. We remove frames from the tracks if the bounding box is too close to the frame border, as these typically indicate a truncated animal.

**Inconsistent Tracks.**   Tracking algorithms may fail when videos contain ambiguous cases or unnatural artifacts. A common failure occurs when multiple animals with similar appearances are present, causing the algorithm to switch identities and track different animals inconsistently. Another failure case arises from video fading effects, which are difficult to detect using shot detection algorithms in earlier stages. In some instances, the tracking algorithm may fail to stop even after a shot change or fade-out, continuing to track a different object or background in the new shot. To mitigate these issues, we apply a threshold on the bounding box IoU between adjacent frames of the same track and remove all frames following a detected low IoU.

**Temporal Postprocessing.**   At this stage, some unqualified frames have been filtered from the tracking results, creating discontinuities. To address this, we apply a post-processing step based on predefined parameters for minimal track length, maximal track length, and the allowed gap within a track. By iterating through all frames in a track, we identify gaps exceeding the allowed threshold or instances where the track reaches the maximal length; in such cases, the subsequent frames are split into a new track. If a gap falls within the allowed threshold, we resegment missing mask silhouette using SAM-2, interpolating bounding boxes from both sides as input prompts. Any tracks shorter than the minimal track length are discarded.

**Object-centric Cropping.**   To obtain the final object-centric video crops for animal reconstruction, we generate square crop boxes centered on the bounding box of the animal in each frame. The size of each crop box is determined by a predefined ratio relative to the mask area. We further apply moving average smoothing to the crop boxes before cropping and resizing all frames to a standardized size. As a final filtering step, we randomly select a cropped RGB image for each track and input it into GPT to identify and remove instances of false detection or heavily occluded animals.

To determine the crop box for each frame in the track, we align the centers of the bounding box and crop box, then extract a square with an area equal to 2× the bounding box area.

We smooth the crop box centers using a moving average with a window size of 10 frames to reduce jitter in the tracking results. Finally, we apply gpt-4o-mini to filter tracks using one sampled frame from each track with the prompt:

> *Does this image show a realistic photo of a {category} without any occlusion? Answer yes or no only.*

### A.5 Features and Auxiliaries Processing

Immediately after each track is saved, we post-process the cropped tracks to generate auxiliary data. Specifically, we integrate ViTPose++ [63] for animal keypoints estimation, DINOv2 [43] for image feature, SEA-RAFT [59] for optical flow estimation, and Depth Anything V2 [67] for depth estimation. For occlusion boundary, we extract depth values at the dilated and eroded mask boundaries, respectively. Then for each pixel on the original mask boundary, we calculate the depth difference between the nearest pixel on the dilated boundary and the nearest pixel on the eroded boundary. This depth difference helps determine whether the pixels outside the animal silhouette belong to the foreground, indicating occlusion, or the background, indicating no occlusion at that region. At this stage, all tracks are stored in the database with computed mean occlusion and optical flow values. If needed, users can further filter the data based on occlusion proportion and optical flow thresholds.

## B  Details of Benchmark

### B.1  Task Formulation

Similar to the 3D animal reconstruction task, which aims to estimate an animal's 3D pose and shape from a single image, the 4D animal reconstruction task seeks to estimate a sequence of 3D poses and shapes from a sequence of frames of the same animal. Beside RGB image input, typical methods also requires other 2D auxiliary data as guidance, such as mask silhouettes and 2D keypoints, obtained either from manual labeling or from pretrained vision models. Formally, given an RGB video input $\mathcal{V}_T = \{v_t\}_{t=1}^T \in \mathbb{R}^{T \times 3 \times H \times W}$ of an animal, along with any required auxiliary input $\mathcal{A}_T = \{a_t\}_{t=1}^T$, in $\{0,1\}^{T \times H \times W}$ in the case of 2D mask or in $\mathbb{R}^{T \times K \times 2}$ in the case of 2D keypoints for instance, a function $f_\theta : \{\mathcal{V}, [\mathcal{A}]\} \mapsto \mathcal{S}$ is expected to output a sequence of posed 3D shapes $\mathcal{S}_T = \{s_t\}_{t=1}^T$ that naturally resembles the shape and pose sequence shown in the input video $\mathcal{V}_T$, where $T$ is number of frames in sequence, $H$ and $W$ are spatial dimensions of the frames, $K$ is number of defined keypoints. Function $f_\theta$ can operate either as a feed-forward model using pretrained parameter $\theta$, or by optimizing $\theta$ at test time. Since different methods operate differently—some requiring large-scale training data [35] while others only perform test-time optimization [74, 7]—our benchmark dataset is designed for evaluation only to ensure fair comparisons.

### B.2  Metrics

**Silhouette Intersection-over-union (IoU).** We follow previous works [9, 76, 4, 58, 49, 69] to employ silhouette intersection-over-union (IoU). Silhouette IoU measures the IoU between the ground truth silhouette mask and the silhouette mask rendered by the reconstructed 3D shape. Although a high 2D IoU does not necessarily correspond to a natural 3D shape due to potential ambiguities, a low IoU reliably indicates that the reconstruction is underperforming.

**Percentage of Correct Keypoint (PCK).**   Following previous works [24, 33, 30, 60, 61, 35], we use the Percentage of Correct Keypoints (PCK) metric, which measures the percentage of projected keypoints that fall within a fixed multiple of a normalizing distance threshold. Studies have defined different distance thresholds. Following [9, 31, 8], we use the square root of the ground-truth mask silhouette area as the normalizing distance threshold.

**Keypoint Transfer (KT).** Since no ground-truth 3D keypoint or shape annotations exist, previous works [61, 35, 29, 30, 69, 33, 8, 28] use Keypoint Transfer (KT) as a proxy for evaluating reconstructed 3D shapes. Specifically, a set of ground-truth 2D keypoints from a source image is projected onto the reconstructed 3D shape surface to establish a mapping with surface vertices. The corresponding vertices are then reprojected from the 3D shape onto a target image with novel view and pose. PCK is computed using the reprojected keypoints and the ground-truth keypoints in the target image. A well-reconstructed 3D shape should exhibit consistency, producing low errors after undergoing this 2D-to-3D-to-2D mapping.

**Mean Per-Joint Velocity Error (MPJVE).** As our work is the first to specifically address the task of 4D animal reconstruction, there are no established metrics for evaluating temporal aspects such as motion smoothness or consistency over time in animal domain. Following related works in human motion estimation [44, 54, 73], we adopt Mean Per-Joint Velocity Error (MPJVE) to quantify the discrepancy in joint velocity within the projected, normalized pixel space. Specifically, for each joint across two consecutive frames, we compute the magnitude of the vector difference between the ground-truth velocity and the predicted velocity. The final MPJVE is obtained by averaging the error over all joints and all frames.

## C  4D-Fauna Implementation Detail

**Preliminary.** 3D-Fauna builds upon MagicPony [61], which learns a prior shape for a specific animal category from diverse images of that category by leveraging self-supervised DINO-ViT [12] features. It then applies instance-specific predicted parameters, such as deformation and articulation, for inverse rendering supervision. Building on this, 3D-Fauna introduces a learnable prior shape bank, which functions as a dictionary of features capable of dynamically combining basis shapes during training and inference to generate diverse instance-specific prior 3D shapes. As a result, 3D-Fauna removes the constraint of training and inference on a single category, enabling it to learn a rich prior shape bank from pan-category images and produce diverse prior shapes at inference time.

### C.1  Canonical Shape Prediction

We use the pretrained 3D-Fauna model as initialization and optimize only the deformation predictor along with the newly introduced per-frame camera and articulation parameters Given a sequence of video frame input $\mathcal{V}_T$, the frozen image encoder will return a sequence of image features $\phi_T$. To ensure a consistent shape across different frames, we compute a mean feature $\bar{\phi} = \frac{1}{T} \sum_{t=1}^{T} \phi_t$, used for predict a prior shape for all frames. We also use the mean feature to input the deformation predictor, guiding it to learn a consistent deformation field that better fit the shape to the masks of the sequence. Specifically, a prior shape predictor $f_{\text{prior}}$ will predict a mesh $(V, F) = f_{\text{prior}}(\bar{\phi})$, where $V$ and $F$ are mesh vertices and faces, and a deformation predictor $f_{\text{deform}}$ will predict a deformation $\Delta V = f_{\text{deform}}(V, \bar{\phi})$, resulting in a canonical shape $(V + \Delta V, F)$.

### C.2  Per Frame Pose Prediction

The inaccurate reconstruction results from 3D-Fauna stem from the fact that its original predictor networks for camera pose and animal articulation are trained on diverse data, making them generalizable but not precise enough for individual instances. To refine the camera pose and animal articulation for a single sequence, we introduce per-frame parameters that are directly optimized, rather than fine-tuning the predictor networks. The outputs from the pretrained networks serve as initialization for this process. Specifically, we introduce per-frame articulation parameters $\{\xi_t\}_{t=1}^{T}$ and camera pose parameters $\{R_t\}_{t=1^T}$. For initialization, $\xi_t = f_{\text{art}}(\phi_t)$ and $R_t = f_{\text{cam}}(\phi_t)$, where $f_{\text{art}}$ and $f_{\text{cam}}$ are pretrained articulation and camera pose predictors, respectively. The canonical shape is first applied with per-frame articulation parameters, following a predefined kinematic tree and skinning function, to transform it into a posed shape. Together with the optimized camera pose parameters and a pretrained texture predictor, the final shape is rendered into an image, mask, and projected keypoints for direct supervision.

### C.3 Keypoints supervision

Since the joints are defined differently between 3D-Fauna framework and output from off-the-shelf keypoint predictor, we identify joints with overlapping defining s for supervision. Specifically, we align node, tail base, and the bottom two joints on four legs, totaling 10 keypoints to calculate keypoint reprojection loss:

$$\mathcal{L}_{\text{kp}} = \|J - \hat{J}\|_2^2 \tag{1}$$

where $J \in \mathbb{R}^{10 \times 2}$ is the ground truth 2D keypoint and $\hat{J} \in \mathbb{R}^{10 \times 2}$ is the predicted keypoints projected onto 2D.

### C.4 Temporal Smoothness Loss

We apply smoothness constraints on articulation angles, posed bones, and camera pose within a batch of frames. Specifically, we minimize both the difference of parameter values between consecutive frames and the difference of changes between consecutive intervals. For smoothness loss on articulation parameter:

$$\mathcal{R}_{\text{smooth,art}} = \sum_{t=1}^{T-1} \|\xi_{t+1} - \xi_t\|_2^2 + \sum_{t=1}^{T-2} \|(\xi_{t+2} - \xi_{t+1}) - (\xi_{t+1} - \xi_t)\|_2^2 \tag{2}$$

The losses are same for posed bone 3D coordinate and camera pose parameters, and overall:

$$\mathcal{R}_{\text{smooth}} = \mathcal{R}_{\text{smooth,art}} + \mathcal{R}_{\text{smooth,bone}} + \mathcal{R}_{\text{smooth,cam}} \tag{3}$$

### C.5 Training Objective

The training objective is essentially the training objective of 3D-Fauna plus the newly added supervisions. From 3D-Fauna:

$$\mathcal{L}_{\text{3D-Fauna}} = \mathcal{L}_{\text{rec}} + \lambda_{\text{hyp}}\mathcal{L}_{\text{hyp}} + \lambda_{\text{adv}}\mathcal{L}_{\text{adv}} + \mathcal{R} \tag{4}$$

where,

$$\mathcal{L}_{\text{rec}} = \lambda_{\text{m}}\mathcal{L}_{\text{m}} + \lambda_{\text{im}}\mathcal{L}_{\text{im}} + \lambda_{\text{feat}}\mathcal{L}_{\text{feat}} \tag{5}$$

and

$$\mathcal{R} = \lambda_{\text{Eik}}\mathcal{R}_{\text{Eik}} + \lambda_{\text{art}}\mathcal{R}_{\text{art}} + \lambda_{\text{def}}\mathcal{R}_{\text{def}} \tag{6}$$

where $\mathcal{L}_{\text{m}}$ is mask reconstruction loss, $\mathcal{L}_{\text{im}}$ is image reconstruction loss, $\mathcal{L}_{\text{feat}}$ is feature reconstruction loss, $\mathcal{L}_{\text{hyp}}$ is viewpoint hypothesis loss, $\mathcal{L}_{\text{adv}}$ is mask shape adversarial loss, $\mathcal{R}_{\text{Eik}}$ is the Eikonal constraint on SDF network for prior shape prediction, $\mathcal{R}_{\text{art}}$ is regularization on articulation parameters, $\mathcal{R}_{\text{def}}$ is regularization on deformations, and $\lambda$s are corresponding balancing loss weights. Adding the new loss terms:

$$\mathcal{L} = \mathcal{L}_{\text{3D-Fauna}} + \lambda_{\text{kp}}\mathcal{L}_{\text{kp}} + \lambda_{\text{kp}}\mathcal{R}_{\text{smooth}} \tag{7}$$

We set $\lambda_{\text{kp}} = \lambda_{\text{smooth}} = 50$. Since our prior shape is fixed, we set $\lambda_{\text{feat}} = \lambda_{Eik} = 0$. All other losses weights follow the implementation in 3D-Fauna.

### C.6 Optimization

We use Adam optimizer with 0.1 learning rate. For each sequence, we construct data into batches of 8 consecutive frames in a sliding window manner. We optimize for 25 epochs for each sequence, starting from the pretrained 3D-Fauna model weights. We run on single L40 GPU with 48 GPU memory.

## D  Dataset Statistics

We show number of frames and number of videos per category of the collected full dataset in Figure 10

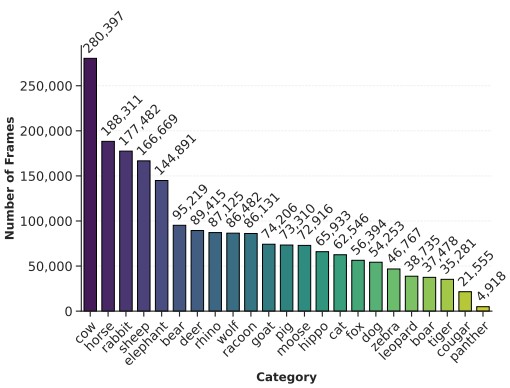

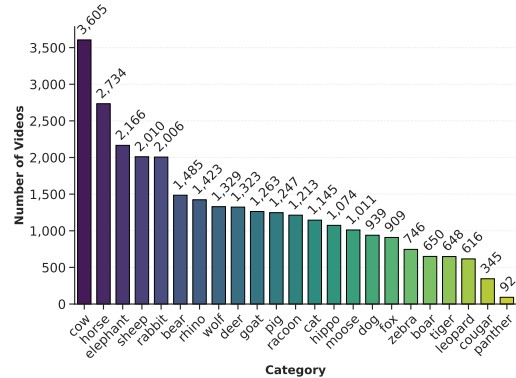

Figure 9: A detailed breakdown of the number of frames collected in full dataset for each animal category.

Figure 10: A detailed breakdown of the number of videos collected in full dataset for each animal category.

# E Motion Type Analysis

We leverage Gemini [55] to annotate each motion video for further motion type analysis. Specifically, we choose a subset of 28 motion type labels defined in AnimalKingdom dataset [41] that are reasonable for quadrupeds. We use gemini-2.5-flash model and let it choose 1-3 motion type labels that best represent the given video. The statistics of the motion type is shown in

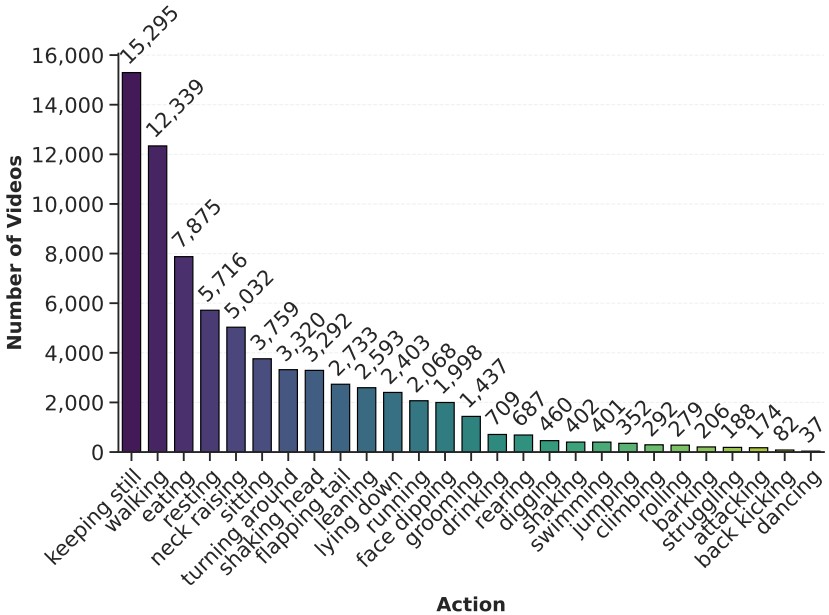

Figure 11: Distribution of motion types across the full dataset. Each video is assigned one to three labels.

# F Visualization of Collected Data

We present a visualization of the video data collected using our proposed data pipeline, consisting of randomly sampled, uncurated object-centric videos with silhouettes applied. More sample data are included in the supplementary materials.

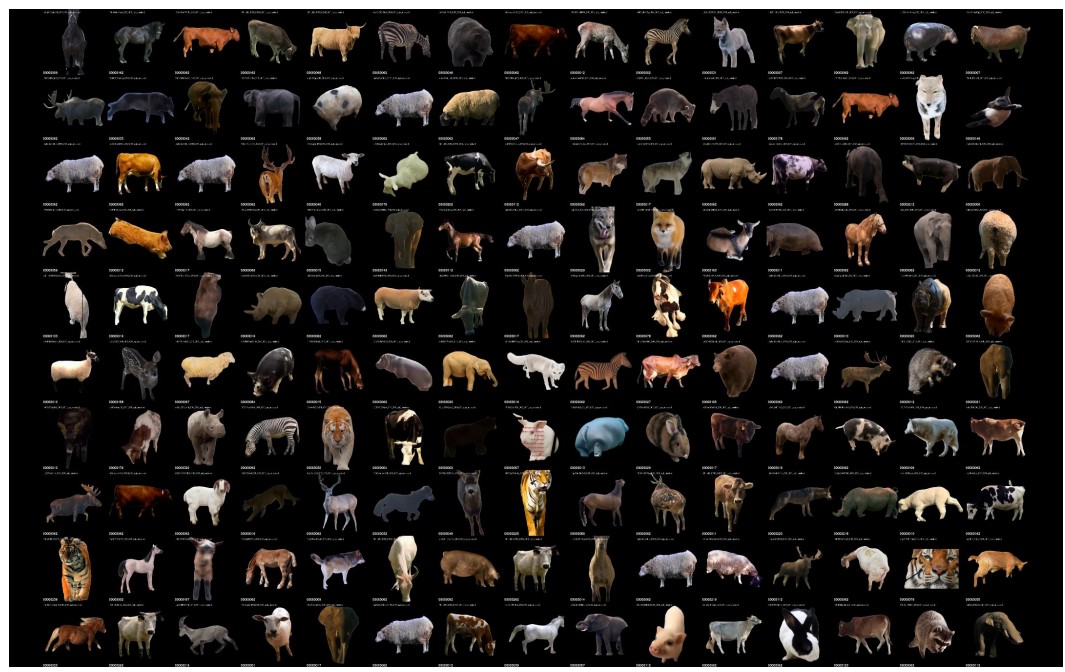

Figure 12: Visualization of random uncurated data collected by out data pipeline.

# G    Visualization of 4D Reconstruction Results

We show some additional visual results of 4D reconstruction results using different methods. Video results on sample data are included in supplementary materials.

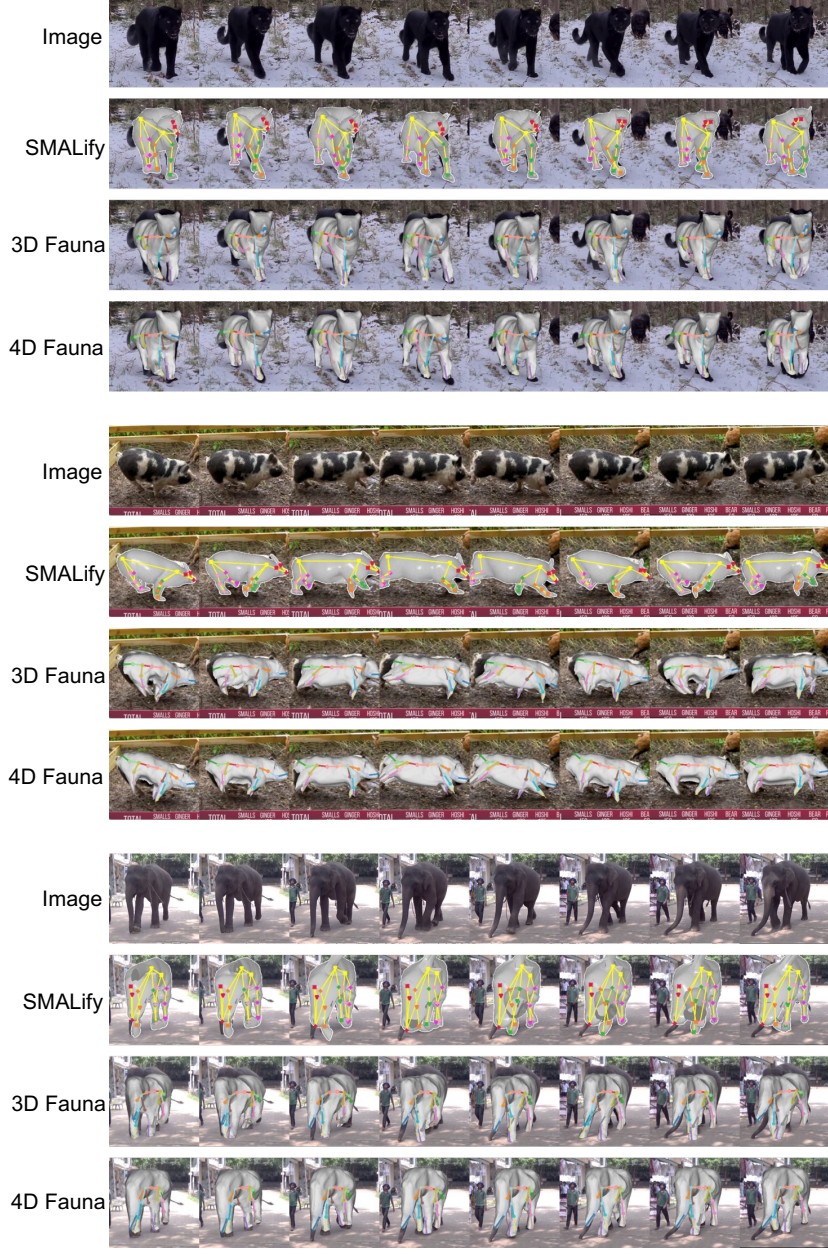

Figure 13: 4D reconstruction results comparison.

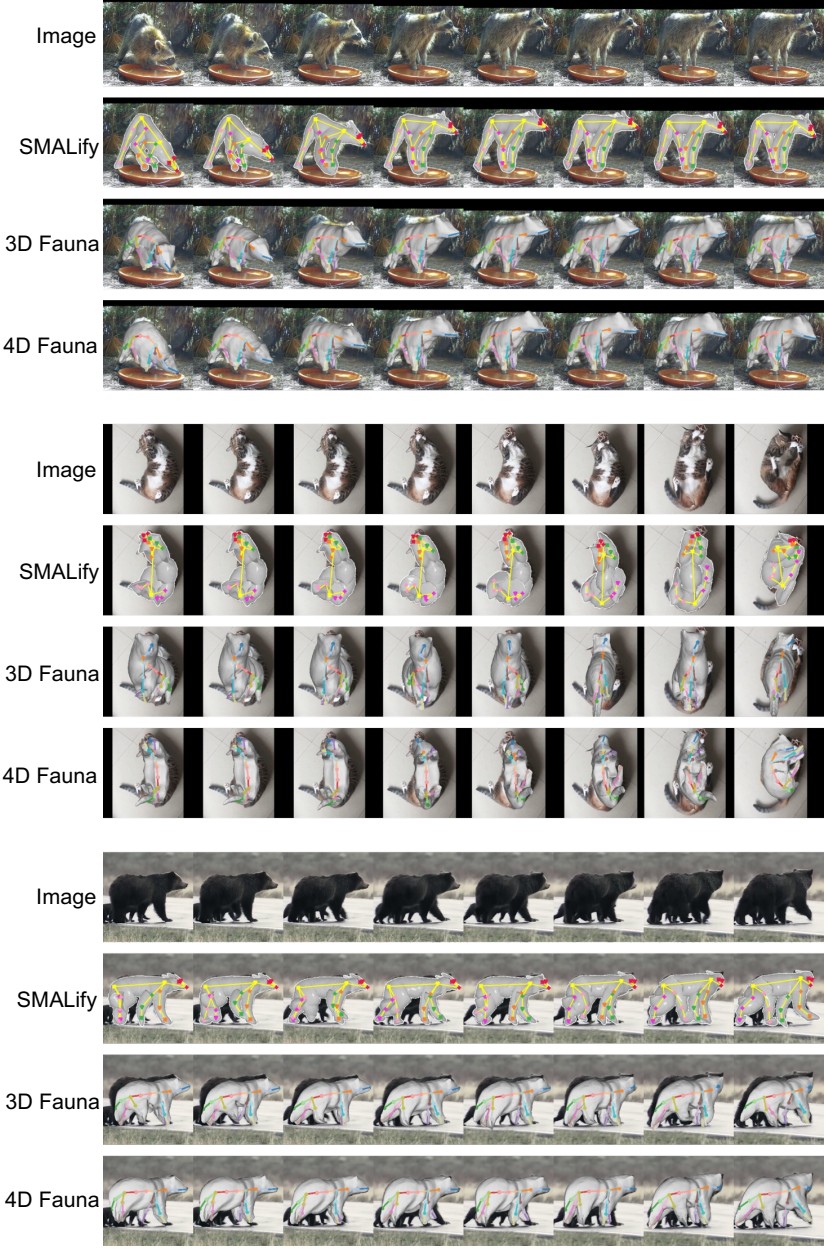

Figure 14: 4D reconstruction results comparison.

