# OpenReview forum: "Web-Scale Collection of Video Data for 4D Animal Reconstruction"
_NeurIPS.cc/2025/Datasets_and_Benchmarks_Track — NeurIPS 2025 Datasets and Benchmarks Track poster_

### Official Review · Reviewer_Wnak · 2025-07-01

**Rating:** 5
**Confidence:** 3

**Summary:**

This paper introduces a fully automated, web-scale pipeline for collecting and preprocessing animal videos from YouTube to support large-scale 4D animal reconstruction tasks. The authors release a 30K-video dataset with 2M frames and propose Animal4D, the first curated benchmark for 4D quadruped pose and shape estimation. In addition, they introduce 4D-Fauna, a model-free baseline built on 3D-Fauna, enhanced with sequence-level optimization for temporal consistency.

**Dataset Code Accessibility:**

Yes

**Ethical Considerations:**

No, there are no or only very minor ethics concerns

**Final Justification:**

This paper presents a scalable pipeline and a large animal video dataset that enables 4D reconstruction at web scale. The proposed benchmark and baseline method are well-motivated and technically sound. While some evaluation metrics and data curation choices have limitations, the authors addressed them reasonably in the rebuttal. Their plan to release the full dataset and expand metrics strengthens the overall contribution.

**Limitations Weaknesses:**

1. The evaluation primarily depends on 2D projection-based metrics, which are known to be weak proxies for assessing 3D reconstruction quality, particularly in capturing depth ambiguities and perceptual realism.

2. Though 23 categories are covered, no insight is given into potential imbalance in motion types, backgrounds, or viewpoints.

3. Animal4D comprises only 10 sequences per category with relatively clean motion and visibility. While useful as a starting point, it lacks "hard" cases such as occlusions, cluttered scenes, or low-light conditions.

**Strengths Contributions:**

1. Proposes a scalable, fully automated pipeline for collecting and processing large-scale in-the-wild animal video data.

2. Develops 4D-Fauna, a model-free baseline that improves temporal coherence through sequence-level optimization.

3. Provides a thorough empirical evaluation, highlighting limitations in current 2D-based metrics for 3D reconstruction.

---

> ### Author Rebuttal · Authors · 2025-07-30
>
> Thank you for the valuable comments and suggestions!
>
> **Q: Limitation of 2D projection-based metrics**
>
> We acknowledge the limitations of current evaluation metrics, particularly those based solely on 2D projections. While these metrics do not perfectly correlate with 3D reconstruction quality, since a high 2D projection score does not necessarily imply a correct 3D shape, a low score often indicates a failure in reconstruction. Despite their limitations, 2D projection metrics remain widely used in the literature, as cited in our paper [6, 65, 2, 49, 42, 59, 19, 28, 25, 51, 52, 30].
>
> To complement this, we also report 3D-aware metrics such as keypoint transfer error, which have become more common in recent works [52, 30, 24, 25, 59, 28, 5, 23]. When 2D projection scores are sufficiently high, keypoint transfer serves as a useful proxy for assessing 3D consistency and typically reflects better overall reconstruction quality.
>
> **Q: More data analysis**
>
> Thank you for the great suggestion! While our primary goal was to demonstrate the scalability of our pipeline across arbitrary animal categories, we recognize that a more detailed breakdown could provide valuable insights for both dataset users and model developers. We will consider including additional analysis and visualizations of these distributional attributes by leveraging off-the-shelf models and pose estimators. This will not only improve transparency but also showcase how our data has the potential to support more downstream tasks like motion classification, viewpoint estimation, etc. We view this as a valuable direction for both improving the dataset's utility and enabling more principled model benchmarking in future work.
>
> **Q: Challenging cases**
>
> Our full dataset consists of nearly 30k video clips, which contain many examples with heavy occlusion, cluttered scenes, as well as other challenging conditions. We will also release the full version of the dataset. From this full dataset, we curate a clean subset for benchmarking purposes only. We will continue to expand the evaluation set to include challenging cases in our future work.

---

> > ### Comment · Reviewer_Wnak · 2025-08-05
> >
> > Thank you for the detailed response. My concerns have been addressed, and I will raise my score accordingly.

---

> > > ### Author Response · Authors · 2025-08-09
> > > **Thank you**
> > >
> > > Thank you again for taking the time to review our paper and provide such valuable feedback. We will incorporate all of your suggestions into our revisions, and please do not hesitate to reach out if you have any further questions or comments.

---

### Official Review · Reviewer_v7EM · 2025-07-02

**Rating:** 4
**Confidence:** 2

**Summary:**

This paper presents a web-scale automated pipeline for collecting and processing animal videos from YouTube, targeting the task of 4D animal reconstruction. The proposed framework enables large-scale, non-invasive, markerless analysis of animal shape and motion, which has traditionally been difficult to achieve at scale. The pipeline comprises the following four stages:

Video scraping: Diverse search queries are generated using GPT and used to automatically download videos from YouTube (Selenium + pytube).

Preprocessing: Videos are segmented into shots using PySceneDetect, and irrelevant clips are filtered using CLIPScore.

Animal tracking: Object-centric cropping and mask tracking are performed using Grounded-SAM-2.

Feature extraction: Keypoints, visual features, optical flow, and depth are extracted using ViTPose++, DINOv2, SEA-RAFT, and Depth Anything V2, respectively.

This pipeline enables the construction of a large-scale dataset with over 30K videos, 2 million frames, and 23 animal categories, with high-quality annotations such as per-frame masks, keypoints, depth, and occlusion boundaries.

From this data, the authors further build Animal4D, a benchmark composed of 230 manually verified sequences (11K frames). They also propose 4D-Fauna, an extension of the existing 3D-Fauna model, incorporating sequence-level optimization via pose smoothness and keypoint supervision. This model serves as the first baseline for 4D animal reconstruction that balances accuracy and perceptual naturalness.

Empirical evaluations show a notable discrepancy: model-based methods (e.g., SMALify) achieve higher 2D metric scores, but model-free methods produce more realistic 3D reconstructions. This gap underscores the need for more 3D-aware evaluation metrics beyond traditional 2D-based measures.

**Dataset Code Accessibility:**

Yes

**Dataset Code Comments:**

The dataset is accessible on Kaggle with valid metadata. Code and instructions are provided, and derived annotations are sufficient for reproducibility. No major issues.

**Ethical Comments:**

The dataset excludes original videos and only shares derived, non-identifiable annotations. No major ethical issues are present.

**Ethical Considerations:**

No, there are no or only very minor ethics concerns

**Final Justification:**

I fully recognise the value of the pipeline and the scale of the dataset. If the camera-ready version were to include quantitative results addressing even one of the points below—for example, adding a numeric multi-view reprojection error or viewpoint-averaged PCK to substantiate viewpoint robustness (Fig. 7)—it would substantially strengthen the empirical case and would likely lead me to raise my score. As things stand, I will keep my rating at 4 (Borderline Accept). I look forward to seeing how this work evolves.

**Limitations Weaknesses:**

While the paper provides thorough evaluations using standard 2D metrics such as Silhouette IoU, PCK, Keypoint Transfer, and MPJVE, these are all projection-based and have inherent limitations when it comes to evaluating the full quality of 3D motion reconstruction. Specifically:

Lack of 3D structural fidelity: These metrics may fail to capture errors in depth or spatial consistency. For example, even if 2D silhouettes and joint positions align, the actual 3D reconstruction may exhibit unnatural shapes—such as crossed limbs or torso distortion—which are not penalized by 2D measures. Figures 7 and 8 illustrate such cases.

Inability to assess motion naturalness and temporal coherence: MPJVE accounts for joint velocity smoothness but does not measure higher-order properties such as motion periodicity, contact consistency, or leg cycle ordering. As a result, temporally smooth but physically implausible motions may receive high scores.

High viewpoint dependence: Since these metrics rely on specific camera views, reconstructions may appear accurate from one perspective but exhibit visible artifacts or unnatural deformations from others.

Given these limitations, more 3D-aware evaluation metrics would strengthen the paper’s claims. Possible directions include metrics based on joint articulation limits, multi-view consistency, mesh-based distances (e.g., Chamfer distance), or perceptual evaluations through user studies. Although the authors acknowledge these issues in the discussion, explicitly incorporating or testing such metrics would further enhance the paper’s impact and robustness.

**Strengths Contributions:**

Clear analysis of the contrast between model-based and model-free approaches
* This paper offers a thoughtful quantitative and qualitative comparison between SMALify (model-based) and 3D/4D-Fauna (model-free). It convincingly demonstrates a discrepancy between evaluation metrics—model-based methods score higher on 2D metrics like IoU and PCK, while model-free methods yield more perceptually natural and realistic 3D shapes and motions.
Timely and important critique of current evaluation practices
* The paper highlights the limitations of widely used 2D metrics in evaluating non-rigid 3D reconstruction and motion. Through empirical evidence and visualizations, it raises a compelling call for more 3D-aware evaluation metrics that better reflect perceptual and structural quality.
Well-balanced new baseline
* The proposed method extends 3D-Fauna with sequence-level optimization and keypoint supervision to improve 2D metrics while preserving naturalness. This balanced design makes it both practically effective and a strong candidate for future benchmark baselines.

---

> ### Author Rebuttal · Authors · 2025-07-30
>
> Thank you for the valuable comments and suggestions!
>
> **Q: Regarding 3D structural fidelity**
>
> We acknowledge the limitations of current evaluation metrics, particularly those based solely on 2D projections. While these metrics do not perfectly correlate with 3D reconstruction quality, since a high 2D projection score does not necessarily imply a correct 3D shape, a low score often indicates a failure in reconstruction. Despite their limitations, 2D projection metrics remain widely used in the literature, as cited in our paper. To complement this, we also report 3D-aware metrics such as keypoint transfer error, which have become more common in recent work. When 2D projection scores are sufficiently high, keypoint transfer serves as a useful proxy for assessing 3D multi-view consistency and typically reflects better overall reconstruction quality.
>
> **Q: Assessing motion naturalness and temporal coherence**
>
> Assessing the naturalness of animal motion is inherently complex due to the vast diversity in animal locomotion. Gaits vary significantly across species and even within a species depending on the context; for instance, a cat’s gait is light and agile, while an elephant’s is slow and heavily constrained. This variability makes it exceptionally difficult to design a universal metric for evaluating properties like motion periodicity across different animals. To our knowledge, no general-purpose metrics that reliably quantify such high-level temporal properties currently exist for the animal domain. This stands in contrast to human motion analysis, where the relative consistency of gaits has enabled more established evaluation methods. While we agree that developing such metrics would be valuable, it remains an open and challenging research problem.
>
> **Q: High viewpoint dependence**
>
> To address viewpoint bias in the evaluation, beyond using keypoint transfer as a proxy to evaluate multiview consistency, we emphasize that qualitative visualization plays a complementary and crucial role in evaluating reconstruction quality. As shown in Figure 7 of our paper, model-based methods such as SMALify can achieve accurate fits in the input view but often produce unnatural deformations or artifacts when visualized from different viewpoints. In contrast, model-free methods tend to produce more consistent and plausible 3D shapes across different viewpoints, largely due to the richer shape priors learned from large-scale data. These qualitative differences are difficult to capture through current metrics alone, and further highlight the need for more viewpoint-robust evaluation strategies. We believe visual inspection across views remains a valuable and necessary component of 3D evaluation.
>
>
> **Q: Why not more 3D metrics**
>
> Regarding 3D evaluation, we agree that evaluation based on accurate 3D data is always a superior choice. However, acquiring high-quality 3D data for animals is extremely challenging and resource-intensive. For instance, datasets described in “VAREN: Very Accurate and Realistic Equine Network” and “The Poses for Equine Research Dataset (PFERD)” involve substantial effort to capture 3D and 4D data for horses in controlled indoor environments. As noted in their descriptions, this process demands a large-scale collaboration between computer vision researchers and animal care specialists.  Despite these efforts, VAREN and PFERD only cover 50 and 5 horses, respectively. In contrast, sourcing from online videos offers the potential to scale to more than millions of individual animals. Moreover, horses are relatively cooperative animals that can be safely handled in structured capture setups. For dangerous or untamed wildlife such as lions or bison, there are currently no practical, safe, or ethical methods to obtain accurate 3D ground truth data beyond passive video recording.
>
> Finally, we would like to emphasize that our main contribution lies in the automatic data pipeline and the large-scale dataset itself. The benchmarking results are provided as a starting point to evaluate current methods and to facilitate further research. One of our goals is to create an open platform that not only supports model comparison but also encourages the community to explore and develop better metrics for evaluating realistic animal motion.

---

> > ### Comment · Reviewer_v7EM · 2025-08-06
> >
> > Thank you for your prompt and thorough rebuttal.
> > Having read the other reviewers’ comments and our ensuing discussion, I fully recognise the value of the pipeline and the scale of the dataset.
> > If the camera-ready version were to include quantitative results addressing even one of the points below—for example, adding a numeric multi-view reprojection error or viewpoint-averaged PCK to substantiate viewpoint robustness (Fig. 7)—it would substantially strengthen the empirical case and would likely lead me to raise my score.
> > As things stand, I will keep my rating at 4 (Borderline Accept). I look forward to seeing how this work evolves.

---

> > > ### Author Response · Authors · 2025-08-09
> > > **Thank you**
> > >
> > > Thank you again for taking the time to review our paper and provide such valuable feedback. We will incorporate all of your suggestions into our revisions, and please do not hesitate to reach out if you have any further questions or comments.

---

### Official Review · Reviewer_BQDF · 2025-07-02

**Rating:** 4
**Confidence:** 5

**Summary:**

In this paper is proposed a valuable and large-scale dataset for animal understanding, including relevant annotations as the silhouette segmentation, 2D pose, optical flow, or boundary tracking to name just a few. To this end, the authors directly consider raw video from youtube, processing them from an object-centric perspective. On balance, 30K clips (with more than 2 million frames) are collected. In addition to that, the authors present 4D-Fauna, a model-free 4D animal reconstruction baseline that adapts previous work to apparently enable more accurate reconstruction on video. An analysis in terms of both model- and data-based approaches is also included in the paper.

**Additional Feedback:**

The qualitative comparisons are hard. Better representations could help the reader.

According to table 1, the claim “while combining the strengths of both traditional model-based and model-free approaches“ seems not to be very effective.

Paraphrasing “This is because model-based methods explicitly optimize pose and shape deformation to align with 2D ground truth“ ok, then, Why don't you provide a comparison in terms of 3D estimates?

**Dataset Code Accessibility:**

Partly

**Dataset Code Comments:**

More 3D annotations should be included for evaluation.

**Ethical Considerations:**

No, there are no or only very minor ethics concerns

**Final Justification:**

After reading the rest of the reviews and the corresponding response, my final score is "borderline accept".

**Limitations Weaknesses:**

While one of the more important claims in the paper is the amount of data in literature to solve the 3D/4D reconstruction problem, I think the authors fail identifying some works where that constraint is not needed, i.e., the battery of approaches where no training data are not considered to infer the solution. At least, I would hope for a discussion of works where the use of animals was the main goal of the paper.

I know, off-the-shelf approaches to infer some relevant features are applied. However, it is worth noting that they are proposed methods in literature, and normally their performance is not 100%. Then, the provided annotations could include a lot of errors, or bad estimations. In my opinion, this should be carefully considered, providing an extra step of supervision. Even SAM-2 can provide incorrect segmentations, for instance.

I disagree with the sentence “it is non-trivial to evaluate a 2D-to-3D lifting task”. I think the community should strive to obtain good 3D data, as was done with other categories a few years ago. Justifying that since there is no data, validation is only in 2D; it is not a suitable evaluation for 3D or 4D.

The set of metrics is well known in literature.

As our work is the first to specifically focus on the task of 4D animal reconstruction, there are no established metrics for evaluating reconstructed motion in the temporal dimension. This is completely false. Some works such as “4DPV: 4D Pet from Videos by Coarse-to-Fine Non-Rigid Radiance Fields” or “Video-specific surface embeddings for articulated 3D shape reconstruction” did it for some categories. This should be properly discussed in the paper.

**Strengths Contributions:**

In general, the paper is well written and it is clear enough.

It is considered a relevant dataset for the community, including a large number of videos for a considerable number of categories. The dataset has value.

---

> ### Author Rebuttal · Authors · 2025-07-30
>
> Thank you for the valuable comments and suggestions!
>
> **Q: Discussion of animal-specific related works**
>
> We agree that some approaches, such as many SMAL-based methods, do not require training data. We have discussed papers specifically about animal reconstruction in the supplementary materials due to space limitation, which we regret were not included in the initial submission:
>
> >The task of 4D animal reconstruction extends from 3D animal reconstruction, which focuses on estimating 3D pose and shape of an animal from a 2D image, and 4D reconstruction results can be obtained by applying 3D reconstruction methods independently to each video frame. Numerous studies have explored the 3d animal reconstruction task, primarily using either a model-based or model-free approach. Typical model-based approaches include ABM [3], SMAL [57], and their variations [6, 7, 25, 34, 35, 43, 58, 59]. These methods start with a prior 3D mesh and optimize a set of predefined pose and deformation parameters to fit the 2D ground truth labels, such as silhouettes and keypoints. These approaches guarantee a consistent 3D shape, however, the predefined shape covers only a limited number of animal categories, and the deformation space is constrained. On the other hand, model-free approaches, including [7, 19, 23, 24, 26, 45, 46, 50, 53], learn a category-specific canonical shape from large-scale image datasets containing different instances of the same category. At test time, they predict instance-specific pose and deformation parameters in a feed-forward manner. Additionally, 3D-Fauna [28] learns a generalizable prior shape bank from pan-category image data, aiming to predict category-agnostic prior shapes at test time. Model-free methods generally accommodate a more diverse range of shapes, however, their feed-forward nature at test time limits the accuracy of shape and pose reconstruction. In this work, we propose a new baseline method that combines the advantages of model-based and model-free approaches by directly optimizing per-frame parameters of a pretrained model-free model at test time, incorporating additional geometric and temporal loss terms on video data.
>
> **Q: Using imperfect off-the-shelf models**
>
> Thank you for the insightful comment. We fully agree that off-the-shelf annotation tools are imperfect and inevitably introduce some noise. However, the primary advantage of our pipeline lies in enabling large-scale data collection without the need for slow and costly manual annotation. At the scale we target, manual labeling would simply be impractical.
>
> We acknowledge the inherent trade-off between data quality and scale. Our approach emphasizes scalability while mitigating quality issues through multiple automated filtering mechanisms. Although some noise remains, the resulting dataset is still highly valuable for many downstream tasks, such as animal motion generation, where data does not need to be 100% correct. This is similar in spirit to large-scale datasets like LAION-5B which, despite relying on automated CLIP-based filtering, have proven effective for training text-to-image generation models.
>
> **Q: Why not use 3D data?**
>
> We agree with the statement that “the community should strive to obtain good 3D data.” However, acquiring high-quality 3D data for animals is extremely challenging and resource-intensive. For instance, datasets described in “VAREN: Very Accurate and Realistic Equine Network” and “The Poses for Equine Research Dataset (PFERD)” involve substantial effort to capture 3D and 4D data for horses in controlled indoor environments. As noted in their descriptions, this process demands a large-scale collaboration between computer vision researchers and animal care specialists.
>
> Despite these efforts, VAREN and PFERD only cover 50 and 5 horses, respectively. In contrast, sourcing from online videos offers the potential to scale to millions of individual animals. Moreover, horses are relatively cooperative animals that can be safely handled in structured capture setups. For dangerous or untamed wildlife such as lions or bison, there are currently no practical, safe, or ethical methods to obtain accurate 3D ground truth data beyond passive video recording.
>
> Given these limitations, our work explores a pragmatic workaround by leveraging large-scale video data, which, while imperfect, provides a feasible and scalable path forward. We also share the hope that future technologies will enable accurate, non-invasive, long-distance 3D capture of animals in their natural habitats.
>
> **Q: Metrics for evaluating temporal motion**
>
> Thanks for suggesting 4DPV and ViSER. Although they are also targeting 4D reconstruction, neither of them actually provides metrics to assess the temporal aspect of motion.
>
> 4DPV: Firstly, the dataset used for quantitative evaluation in this work includes 3D mesh ground truth, which is required to compute the Chamfer Distance and F-score. Moreover, their reported scores are simply averaged over all frames; therefore, they do not capture temporal aspects of motion such as smoothness, velocity, or jitter.
>
> Similarly, ViSER also reports keypoint transfer error averaged across all frame pairs, which is the same protocol we follow. While useful for evaluating spatial consistency, this metric also does not directly assess temporal quality.
>
> Evaluating the quality of estimated animal motions is challenging. If the reviewer could suggest any metrics that can better assess the quality of estimated temporal animal motions, we would be happy to incorporate more evaluations in the final version of the paper. We will also modify our wording and include a discussion about these works in the revised version.
>
> **Q: More qualitative results**
>
> Again we apologize for not including the supplementary material in the initial submission, which has more visualization available.  As we are also not allowed to provide additional visualizations during the rebuttal, we will include more visualizations in the revised version and will provide 3D animations of the reconstruction results on our project page. All code, data, and models will be released publicly to facilitate future work.
>
> **Q: Table 1 explanation**
>
> Model-based approaches such as SMAL, which rely on projection-based optimization, are typically effective at fitting 2D silhouettes and keypoints. However, they often neglect the overall plausibility of the resulting 3D shape and pose due to their limited shape space and reliance on 2D cues. In contrast, model-free methods like 3D-Fauna generate 3D meshes via feed-forward prediction, producing more naturally plausible shapes by leveraging large-scale training data, but they tend to be less accurate in 2D projections of shape and keypoints.
>
> Our claim of “combining the strengths of both traditional model-based and model-free approaches” simply refers to leveraging the naturally plausible 3D outputs of model-free methods, and then refining them through projection-based optimization similar to model-based techniques. This hybrid approach aims to achieve both 3D plausibility and improved 2D consistency.
>
> We acknowledge the limitations of current evaluation metrics, particularly those based solely on 2D projections. While the numerical results may not definitively show that our method outperforms both baselines, qualitative results demonstrate more natural 3D reconstructions and improved alignment with 2D observations, supporting our claim.
>
> Finally, we would like to emphasize that our primary contribution lies in the data pipeline and the large-scale 2D video dataset. Our benchmark enables evaluation of various approaches under realistic and challenging conditions, and we believe it opens substantial opportunities for future improvement in this area.
>
> **Q: Why not provide 3D evaluation?**
>
> We reiterate that collecting accurate 3D data is expensive and logistically challenging, which is why our work explores an alternative, more scalable 2D-based pathway. While we agree that the community should continue to pursue improved methods for acquiring high-quality 3D data, we believe the 2D-based approach remains highly valuable, particularly for scaling to diverse, in-the-wild scenarios where 3D capture is currently impractical.

---

> > ### Comment · Reviewer_BQDF · 2025-08-07
> >
> > Thank you very much for the response. In general, the authors addressed all my comments!
> >
> > Regarding the question: Discussion of animal-specific related works
> >
> > Well, I agree with the answer, but as I commented, other works can solve that without assuming "a prior 3D mesh", as the authors indicate. Then, to be fair, in my opinion, the authors should consider all lines of research, especially if they do not embrace the strongest priors discussed. I found several works along these lines, perhaps the most recent being this: "Safari from Visual Signals: Recovering Volumetric 3D Shapes". This is an instance where the acquisition of 3D data is hard or directly impossible. To strengthen the ideas presented in this paper and help the reader, I believe that all lines should be carefully discussed.
> >
> > "Metrics for evaluating temporal motion". I am not entirely sure about this answer. Both methods are capable of obtaining a 4D reconstruction; in fact, temporal constraints are imposed. Consequently, I think it's fairly straightforward to obtain a temporal motion, especially when an averaged score over all frames is provided.
> >
> > Again, thanks again for providing such a convincing response to all the reviewers.

---

> > > ### Author Response · Authors · 2025-08-07
> > >
> > > Thank you for the further comments!
> > >
> > > We will extend our discussion on related works to cover a broader range of animal reconstruction paradigms, including “Safari from Visual Signals”, which represents a different and interesting approach.
> > >
> > > Regarding temporal metrics, our main point is that, to the best of our knowledge, there exist no well-established metrics for evaluating the temporal quality of 4D reconstruction. While both methods impose temporal constraints and produce 4D reconstructions, they do not introduce quantitative metrics that are genuinely sensitive to temporal inconsistencies. In our evaluation, we follow their protocol by reporting scores averaged over all frames. That said, we will broaden our discussion to include these 4D reconstruction methods and provide both qualitative and quantitative comparisons.

---

> > > ### Author Response · Authors · 2025-08-08
> > >
> > > We want to thank you once again for taking the time to review our paper. As the discussion will end today, we would like to confirm if we have addressed all your concerns. If you have any other questions, please let us know and we will address them as soon as possible. If you are satisfied with our response, we would greatly appreciate it if you could consider updating your final rating accordingly. Thank you!

---

### Official Review · Reviewer_BUq6 · 2025-07-03

**Rating:** 5
**Confidence:** 5

**Summary:**

This paper introduces Animal4D, a large-scale dataset and benchmark for 4D animal reconstruction (3D shape + motion over time). The authors develop an automated pipeline to scrape and process YouTube videos into 30K object-centric clips (2M frames) with annotations (masks, keypoints, optical flow, etc.). They curate a high-quality subset, Animal4D Benchmark (230 sequences, 11K frames), for evaluating 4D reconstruction. The paper also proposes 4D-Fauna, an enhanced baseline method that adapts a model-free approach with temporal optimization, improving reconstruction accuracy and naturalness. Key contributions include the scalable data engine, the benchmark, and analysis exposing gaps in current evaluation metrics.

**Dataset Code Accessibility:**

Partly

**Dataset Code Comments:**

The authors release part of the data currently, which are impressive. The unreleased parts are Youtube videos, which could not be re-published due to Youtube's Terms of Service. Other derived data are fully released.

**Ethical Considerations:**

No, there are no or only very minor ethics concerns

**Final Justification:**

After reading all reviews and rebuttals, I still maintain my initial rating.

1. The rebuttal clearly answers my questions.

2. Reviewer BQDF proposes questions about true 3d annotations. In my mind, aquiring 3D data involves multi-view recording or laser scanning, which isstill hard. Video data itself has the potential to help improve 4D intelligence.

**Limitations Weaknesses:**

1. Missing reference:
[1] AniMer: Animal Pose and Shape Estimation Using Family Aware Transformer, CVPR 2025.  At L. 30, while summarizing sota prior-based (SMAL-based) methods, [1] deserves to be mentioned.
[2] EgoPet: Egomotion and Interaction Data from an Animal's Perspective, ECCV 2024. At "Related Works" section, while summarizing animal video dataset, I think EgoPet is not negligible. It records several animal motions from animals' view.

2. Table.1 only involves optimization-based SMALify for model-based baseline, lacks the comparison with end-to-end learning based methods. Learning baed methods would produce less unnatural poses than SMALify. In the future, incorporating learning-based methods as baselines (such as BITE, AniMer, BARC, etc.) would make the paper stronger.

3. The scope is a bit narrow, that it only involves quadrupeds. Future work may extend to other animal categories (birds, marine life, etc.)

**Strengths Contributions:**

1. The data collection pipeline is scalable. And the dataset presented shows an order of magnitude (30K videos vs. prior 2.4K clips) improvement, enabling large-scale animal motion analysis.

2. Animal4D Benchmark is carefully curated (human-validated masks/keypoints) and includes diverse quadruped motions, filling a critical gap for 4D tasks.

3. The improvement of 4D-fauna over 3D-fauna is meaningful.

4. Sec. 6 carefully states the existing limitations of current pipeline, pointing out the future direction.

---

> ### Author Rebuttal · Authors · 2025-07-30
>
> Thank you for the valuable comments and suggestions!
>
> **Q: Reference of AniMer and EgoPet**
>
> We provide comparison results against AniMer below and will add more discussion in the final version. We also agree that EgoPet is an interesting paper that is worth discussing, and we will reference it as well.
>
> **Q: More learning-based baselines**
>
> Thanks for the suggestion. We present the updated table with AniMer results and will incorporate additional baselines in the revised version.
> | Method | IoU↑ | PCK @0.1↑ | PCK @0.05↑ | KT-PCK @0.1↑ | KT-PCK @0.05↑ | MPJVE↓ |
> |---|---|---|---|---|---|---|
> | SMALify (OPT, MB) | 0.867 | 0.954 | 0.787 | 0.623 | 0.372 | 0.023 |
> | AniMer (FF, MB) | 0.677 | 0.537 | 0.199 | 0.566 | 0.256 | 0.038 |
> | 3D-Fauna (FF, MF) | 0.670 | 0.470 | 0.177 | 0.329 | 0.130 | 0.058 |
> | 4D-Fauna (FF+OPT, MF) | 0.814 | 0.664 | 0.317 | 0.418 | 0.193 | 0.044 |
>
> OPT: optimization-based method,
> FF: Feedforward method
>
> MB: Model-based method,
> MF: Model-free method
>
> **Q: Extending beyond quadrupeds**
>
> We agree this is a limitation of the existing 3D/4D animal reconstruction methods, which rely either on a pre-defined template like SMAL or a given quadrupedal skeleton. We are currently working on extending our method to broader categories and looking forward to sharing our work in the future.

---

> > ### Comment · Reviewer_BUq6 · 2025-08-05
> > **Post rebuttal rating: 5: Accept**
> >
> > This is the post-rebuttal comment. I choose to maintain my initial rating, and wish the full dataset would be available soon.
> >
> > Still, I have some further open questions (not affect my rating).
> >
> > 1. Why SMALify performs the best in the table? As an optimization method without temporal refinement, it would frequently bring temporal jittering and false alignment when some body parts are occluded (e.g. a dog is siting, or the animal is truncated by the image border). A video comparison may help an intuitive understanding.
> >
> > 2. Maybe one of the reasons for question 1 is that in most cases of the dataset, animals are standing or walking ( from the downloaded dataset samples). A clarification on the pose diversity or pose distribution may better position this dataset in the 4D animal area.

---

> > > ### Author Response · Authors · 2025-08-05
> > >
> > > Thank you for the further comments. We are currently preparing more visualizations of the dataset and finalizing the documentation. We will be ready to release the dataset before the camera-ready deadline. To answer the two questions:
> > >
> > > 1. The strong performance of SMALify is largely due to the limitations of current projection-based evaluation metrics, which align closely with the optimization objectives of non-learning model-based methods like SMALify. Since SMALify explicitly fits pose and shape to match 2D ground truth silhouettes and keypoints, it naturally achieves high scores on metrics such as IoU, PCK, and MPJVE.
> > > For 3D-aware metrics like KT-PCK, although SMALify may produce unnatural shapes or poses, its performance remains high for two main reasons: (1) it still produces plausible reconstructions in many cases, and (2) many video sequences lack significant viewpoint changes, limiting the ability of keypoint transfer to penalize unrealistic shapes and poses.
> > > Therefore, we emphasize that qualitative visualization plays a complementary and crucial role in evaluating reconstruction quality when currently there are still no perfect metrics for this task.
> > >
> > >
> > > 2. We agree with this observation. As noted above, the lack of significant viewpoint variation within individual videos can contribute to higher performance on existing metrics. We are currently preparing more analysis of the datasets, visualizing the distributions of attributes such as motion type and viewpoint. We will include these visualizations in the final release of the dataset.

---

> > > ### Author Response · Authors · 2025-08-09
> > > **Thank you**
> > >
> > > Thank you again for taking the time to review our paper and provide such valuable feedback. We will incorporate all of your suggestions into our revisions, and please do not hesitate to reach out if you have any further questions or comments.

---

### Note · Authors · 2025-08-12

We would like to sincerely thank all the reviewers and ACs for their valuable feedback, constructive discussion, and efforts in handling the review process for our paper. In summary, following the discussion period, all four reviewers expressed satisfaction with our responses and appeared to be positive about the submission. One reviewer (BQDF) noted that “the authors addressed all my comments!”, but may not have had the time to update the final review and rating.

Again, we appreciate the helpful comments and feedback from the reviewers and AC and will ensure the incorporation of the rebuttal into the revision.

---

### Decision · Program_Chairs · 2025-09-18

**Decision:**

Accept (poster)

**Comment:**

This paper presents a valuable advancement for 4D animal reconstruction research by delivering a large-scale Animal4D dataset, a well-curated benchmark, and a strong baseline method 4D-Fauna. While some evaluation metrics and data curation choices have limitations, the authors have been receptive and responsive to reviewer feedback, addressing concerns and clarifying methodological choices in the rebuttal and during the author-reviewers discussion. This results in all the reviewers recommending acceptance in the final rating.

Given the authors’ commitment to incorporating further metrics and visual materials in the final version, the work merits acceptance. The AC is happy to accept the paper. Congratulations! Please be aware that the authors are strongly encouraged to address the outlined limitations and integrate enhanced empirical evidence in the camera-ready version.